# Raman Monitoring of *Staphylococcus aureus* Osteomyelitis: Microbial Pathogenesis and Bone Immune Response

**DOI:** 10.3390/ijms26178572

**Published:** 2025-09-03

**Authors:** Shun Fujii, Naoyuki Horie, Saki Ikegami, Hayata Imamura, Wenliang Zhu, Hiroshi Ikegaya, Osam Mazda, Giuseppe Pezzotti, Kenji Takahashi

**Affiliations:** 1Department of Orthopaedics, Graduate School of Medical Science, Kyoto Prefectural University of Medicine, Kamigyo-ku, 465 Kajii-cho, Kyoto 602-8566, Japan; sfujii03@koto.kpu-m.ac.jp (S.F.); n-horie@koto.kpu-m.ac.jp (N.H.); 2Ceramic Physics Laboratory, Kyoto Institute of Technology, Sakyo-ku, Matsugasaki, Kyoto 606-8585, Japan; s.ikegami1771@gmail.com (S.I.); hyt8888@outlook.jp (H.I.); wlzhu@kit.ac.jp (W.Z.); 3Biomedical Engineering Center, Kansai Medical University, 1-9-11 Shin-machi, Hirakata 573-1191, Japan; 4Department of Dental Medicine, Graduate School of Medical Science, Kyoto Prefectural University of Medicine, Kamigyo-ku, 465 Kajii-cho, Kyoto 602-8566, Japan; 5Department of Forensics Medicine, Graduate School of Medical Science, Kyoto Prefectural University of Medicine, Kamigyo-ku, 465 Kajii-cho, Kyoto 602-8566, Japan; ikegaya@koto.kpu-m.ac.jp; 6Department of Immunology, Graduate School of Medical Science, Kyoto Prefectural University of Medicine, Kamigyo-ku, 465 Kajii-cho, Kyoto 602-8566, Japan; mazda@koto.kpu-m.ac.jp; 7Department of Orthopedic Surgery, Tokyo Medical University, 6-7-1 Nishi-Shinjuku, Shinjuku-ku, Tokyo 160-0023, Japan; 8Biomarker Disease Laboratory, IRCCS San Camillo Hospital, Via Alberoni 70, 30126 Venice Lido, Italy; 9Department of Molecular Science and Nanosystems, Ca’ Foscari University of Venice, Via Torino 155, 30172 Venice, Italy

**Keywords:** bone infection, *Staphylococcus aureus*, Wistar rat model of osteomyelitis, Raman spectroscopy, host immunological response

## Abstract

*Staphylococcus aureus* is the most common pathogen causing osteomyelitis, a hardly recoverable bone infection that generates significant burden to patients. Osteomyelitis mouse models have long and successfully served to provide phenomenological insights into both pathogenesis and host response. However, direct in situ monitoring of bone microbial pathogenesis and immune response at the cellular level is still conspicuously missing in the published literature. Here, we update a standard pyogenic osteomyelitis in Wistar rat model, in order to investigate bacterial localization and immune response in osteomyelitis of rat tibia upon adding in situ analyses by spectrally resolved Raman spectroscopy. Raman experiments were performed one and five weeks post infections upon increasing the initial dose of bacterial inoculation in rat tibia. Label-free in situ Raman spectroscopy clearly revealed the presence of *Staphylococcus aureus* through exploiting peculiar signals from characteristic carotenoid staphyloxanthin molecules. Data were collected as a function of both initial bacteria inoculation dose and location along the tibia. Such strong Raman signals, which relate to single and double bonds in the carbon chain backbone of carotenoids, served as efficient bacterial markers even at low levels of infection. We could also detect strong Raman signals from cytochrome *c* (and its oxidized form) from bone cells in response to infection and inflammatory paths. Although initial inoculation was restricted to a single location close to the medial condyle, bacteria spread along the entire bone down to the medial malleolus, independent of initial infection dose. Raman spectroscopic characterizations comprehensively and quantitatively revealed the metabolic state of bacteria through specific spectroscopic biomarkers linked to the length of staphyloxanthin carbon chain backbone. Moreover, the physiological response of eukaryotic cells could be quantified through monitoring the level of oxidation of mitochondrial cytochrome *c*, which featured the relative intensity of the 1644 cm^−1^ signal peculiar to the oxidized molecules with respect to its pyrrole ring-breathing signal at 750 cm^−1^, according to the previously published literature. In conclusion, we present here a novel Raman spectroscopic approach indexing bacterial concentration and immune response in bone tissue. This new approach enables locating and characterizing in situ bone infections, inflammatory host tissue reactions, and bacterial resistance/adaptation.

## 1. Introduction

Orthopedic surgeons can exploit several tools and techniques to detect infections in bones before, during, and after surgery [1]. Preoperative imaging, such as X-rays, high-resolution computerized tomography (CT) scans, or magnetic resonance imaging, can help to preliminarily identify areas of infection [2,3,4]. During surgery, the surgeon visually assesses the condition of the bone through judging the level of necrosis and the abnormality of its shape. Intraoperative imaging-guided biopsy using CT fluoroscopy is the preferred initial procedure for diagnosing osteomyelitis [5], while samples of bone tissue or fluid can eventually be collected during surgery and sent for culture and sensitivity testing [6]. Sonication is an additional supporting tool, where samples of bone extracted from patients are placed in a solution and exposed to ultrasound to release bacteria for analysis [7,8]. This set of well-established procedures generally enables surgeons to identify the severity of infection and the specific bacteria causing it, thus guiding surgical decisions effectively. However, pyogenic osteomyelitis is one of the most difficult bone diseases to treat in the field of orthopedics. Debridement of infected tissue is the standard procedure in treating suppurative osteomyelitis, but it is generally difficult to visually distinguish normal tissue from infected tissue during surgery because infection is hidden in the depth of bone tissue. As a consequence, its partial elimination might lead to the need for successive surgery with additional burden on the patient.

Raman spectroscopy has been widely used to construct quantitative procedures capable of assessing bone structure, quality, and infections [9,10,11]. This vibrational spectroscopic technique allows for the analysis of bone tissue at the molecular level, providing insights into the presence of bacterial pathogens and revealing biochemical changes associated with infection. Specific bacterial species can also be identified nearly in real time based on unique spectral signatures [12,13], thus aiding in the diagnosis of the cause of osteomyelitis and necrosis [14]. Since infected bone tissue often undergoes changes in mineral composition and organic matrix structure, Raman spectroscopy can unfold such changes [10], helping clinicians to assess in real time the severity of the infection [15,16]. By this same approach, changes in the spectral profile over time can be used to monitor the effectiveness of drug treatments against bone pathogens [17]. Being a non-invasive method, application of Raman spectroscopy during surgery could minimize removal of surrounding (healthy) tissue, making it a valuable tool in clinical settings [18]. Fiber Raman devices could thus be used to detect bone infections during surgery, since they offer the advantage of being minimally invasive and label-free and can provide real-time feedback during surgical procedures. The possibility of obtaining immediate results in the surgery room could help surgeons make informed decisions about debridement and the need for specific antibiotic treatment. However, while showing promise in diagnostics and monitoring of bone infections, more research is needed to fully establish clinical applications of Raman spectroscopy in the field of orthopedics.

Building upon previous Raman studies on bone infections [13,14,15,16,19,20,21], herein, we demonstrate a Raman spectroscopic workflow for interrogating the carotenoid staphyloxanthin molecules of a *Staphylococcus aureus* osteomyelitis Wistar rat model [22], given the important role of these molecules as markers for this bacterium and its physiological stress state. First, reference spectra from elementary molecules were searched for in the literature or newly collected to identify specific markers for bacteria, osteoblasts, and bone immunochemistry in infected rat tibias. Then, the Raman intensities and spectral shifts in staphyloxanthin molecules were collected for different bacterial inoculation doses, at different regions of bone as a function of both distance from inoculation site and probe depth in bone tissue. Moreover, we could collect Raman signals from cytochrome *c* molecules and their oxidized form, which unfolded bone cells’ immunoreaction as a function of infection level. With the goal of unfolding osteomyelitis pathology, the development of functional spectroscopic parameters linking molecular and cellular interactions between bone tissue and bacteria lays the foundations for an advanced multiomic insight into the pathology of bone infections.

## 2. Results

### 2.1. Osteomyelitis Rat Model and Microscopy Observations

We extracted normal and bacteria-inoculated rat tibias at 7 and 35 d post inoculation. Figure 1a,b show a picture of an anesthetized rat after inoculation and its extracted tibia after 7 d post inoculation, respectively. Figure 1c–f show pictures of rats at 7 days post injection with 100 μL PBS only, and with the same amount of PBS added with 10^3^, 10^5^, and 10^8^ CFU/mmL concentrations of *S. aureus* bacteria, respectively. The same procedure was followed at 35 d post bacteria inoculation. No definite swelling was found in any of the inoculated tibias, except for the tibia inoculated with PBS only (cf. Figure 1c). No definite sign of inflammation could be found in any of the extracted tibias based on external visual inspections. However, bacterial colonies could be clearly identified within the bone marrow of sectioned tibias upon microscopy inspections. Figure 2a and the enlarged inset in it (Figure 2b) clearly show such bacterial colonies as seen for the majority of locations at 7 d post infection, namely, with no clear sign of abscess. However, formation of abscesses could sporadically be observed even at 7 d post infection (Figure 1c). The meaning of the above microscopy observations will be extensively discussed in the forthcoming Section 3.1, Section 3.2 and Section 3.3.

### 2.2. Reference Raman Spectra and Probe Depth Assessment

Figure 3a,b show reference spectra preliminarily collected with focus on the surface of a healthy tibia and *S. aureus* bacteria cultured in vitro, respectively. The bone spectrum was dominated by the signal located at 961 cm^−1^, which can be assigned to the symmetric stretching of the P–O bonds of the PO_4_^3−^ tetrahedron of hydroxyapatite [23]. The sharpness of this band is a label of the crystalline character of the hydroxyapatite structure in cortical bone [24,25,26,27], which represents the only tissue present in the probe when focused on the bone surface. Additional bands related to PO_4_^3−^ tetrahedron vibrations can be found at around 460 cm^−1^ (O–P–O symmetric bending), 590 cm^−1^ (O–P–O asymmetric bending), and 1060 cm^−1^ (P–O asymmetric stretching (cf. red arrows and labels in inset to Figure 3a). The remaining bands mainly arise from molecular vibrations related to the collagen structure [28]. We used here the intensity of the P–O stretching signal of hydroxyapatite at 961 cm^−1^ to calibrate the penetration of the green laser beam within the bone under the optical probe conditions selected for this study (i.e., 20× objective lens and 10 mW laser power). Figure 3b shows the results of a defocusing calibration of the optical probe performed by shifting the position of its focal plane along the in-depth focal axis, *z* (cf. draft and labels in inset) [29]. The plot of Raman intensity as a function of the focal abscissa above (−*z*) and below (+*z*) the sample surface reveals that the present Raman probe could retrieve information down to ~500 μm below its focal plane (~15% of the total signal). These calibration data suggested that, over a total rat tibia thickness of approximately 3~5 mm (cf. Figure 1b), a suitable approach could be to shift the focal plane below the bone surface in order to reach buried portions of bone marrow.

Although laser wavelength/power and the choice of objective lens should be tailored case-by-case depending on the type of bone sample to be investigated, the results in Figure 3b suggest that bone tissue possesses a level of transparency enabling non-destructive evaluations of its internal structure both in health and disease. Further evidence supporting this assertion will be given with the analysis of osteomyelitic bones shown in the next section. Two main bands at 1176 and 1535 cm^−1^ dominate the spectrum of in vitro cultured *S. aureus*, as shown in Figure 3c. These two Raman signals arise from C–C and C=C stretching, respectively, and their strong enhancement as compared to other molecular markers of the bacterial structure is due to the abundance of the above two bonds in the carotenoid chain of staphyloxanthin [30]. The structure and the Raman spectrum of the staphyloxanthin molecule are shown in Figure 4a and Figure 4b, respectively. Stoichiometric staphyloxanthin consists of a β–D–glucopyranose ring in which the hydroxy groups at position 1 and 6 have been acylated by a carotenoid chain and a 12–methyltetradecanoyl group, respectively (cf. labels in inset to Figure 4a). In its stoichiometric form, the staphyloxanthin molecule contains *N* = 10 C=C bonds. In the Raman spectrum of pure staphyloxanthin compound (Figure 4b) [31], both C–C and C=C stretching bands are seen at lower wavenumbers as compared to the spectrum of *S. aureus* bacteria (cf. spectra in Figure 3c and Figure 4b). These spectral-shift differences have a specific meaning related to the non-stoichiometric form of the molecule, as described in a forthcoming section. Two additional bands in the staphyloxanthin spectrum, which are seen at around 1018 and 1423 cm^−1^, could be assigned to C–O stretching and C–H deformation, respectively (cf. Figure 4b). As can be seen from comparing Figure 3a,c, the prominent Raman markers of staphyloxanthin and hydroxyapatite do not overlap each other, thus allowing their clear differentiation in the analysis of osteomyelitic bone. The C–C and C=C carotenoid Raman markers provide insight into biochemical factors that have been used to differentiate methicillin-sensitive from methicillin-resistant *S. aureus* strains, as well as the development of mycoplasma pneumoniae [30,32]. In the remainder of this paper, we shall show that these markers can also be used to assess the degree of pathogenicity of the osteomyelitic infection through revealing the degree of oxidation of the staphyloxanthin chain.

Another Raman feature of interest in the present context is the haem ring of the cytochrome *c* molecule. Cytochrome *c* is a hemeprotein, which, after being released from the mitochondrion into the cytosol, becomes oxidized and results in the disassembly of cells’ proteins as the ultimate sign of apoptosis [33]. Despite the variety and complexity of the mechanisms involved with cells’ apoptosis, monitoring the Raman signals from cytochrome *c* gives a straightforward tool to generally characterize the apoptotic attitude of living cells [13]. Structures of cytochrome *c* in reduced and oxidized states are shown in Figure 5a,b, respectively, and their respective Raman spectra in the wavenumber interval 600~1750 cm^−1^ are displayed in Figure 5c [34]. In the spectrum of cytochrome *c* in reduced state (haem Fe^2+^), three main signals can be located at (cf. Figure 5c): 750 cm^−1^ (pyrrole ring-breathing), 1136 cm^−1^ (C–CH_3_ stretching), and 1596 cm^−1^ (C–C and C–CH stretching) [35]. Upon oxidation (haem Fe^3+^), two additional bands appear at 1564 and 1644 cm^−1^, which can be assigned to asymmetric C–C stretching and C=C/C=N stretching, respectively [35]. The appearance of these additional bands, in particular the one at 1644 cm^−1^, can be exploited to assess the level of haem oxidation and, thus the fraction of apoptotic cells [36,37,38,39,40]. This unique spectroscopic circumstance will be exploited in the present work to characterize osteomyelitic bone; more details about this point will be given in the forthcoming Section 3.

### 2.3. Raman Spectra of Bone at the Acute Stage of Osteomyelitis

Figure 6a shows average Raman spectra as collected in the neighborhood of the inoculation point of rat tibias immediately after extraction at 7 d post bacterial injection. From top to bottom, the spectra refer to samples inoculated with 100 μL PBS containing 10^8^, 10^5^, 10^3^ CFU/mmL bacterial concentrations, samples inoculated with only 100 μL PBS, and non-inoculated healthy bone as control samples (number of measurements, *n* = 6 for each rat sample; cf. labels in inset). The laser spot size in the focal plane was ~5 μm, while the probe broadened inside the bone tissue with collecting detectable Raman scatters down to 500 μm below the focal plane (although with increasingly weaker intensity; cf. Figure 3b). In order to record Raman spectra as close as possible to the point of bacterial injection (but yet compatibly with sufficient scattering intensity), we defocused the probe by ~300 μm into the bone subsurface. Considering that the total thickness of rat tibia and its cortical bone thickness are 3~5 mm and ~700 μm, respectively [41,42], the selected probe size and focal depth enabled us to partly include in the measurements cancellous bone and bone marrow zones below the cortical bone structure. These buried zones are indeed the main locations hosting bacterial infection. Considering the probe shape as approximately that of a truncated cone with an upper circle in the focal plane with surface area ~20 μm^2^ and a lower circle at a depth of ~500 μm with surface area of ~0.2 mm^2^, the total volume probed in a single measurement can be estimated as ~3.3 × 10^7^ μm^3^. Despite the fact that the intensity of signals scattered from deeper zones is weaker, the volume of the adopted probe ensured that a large number of bacteria and bone cells (sizes of 0.5~1.5 and 10~50 μm, respectively) were included in the probe at each measurement. In addition, we collected six measurements at slightly shifted locations around the selected zone and computed average spectra in order to ensure statistical validity in the spectroscopic assessments.

As seen, in all bacteria-inoculated spectra, main signals belonging to staphyloxanthin and cytochrome *c* (as described in the previous section) could be clearly detected together with the P–O stretching band of bone hydroxyapatite (at 961 cm^−1^), which was used for normalizing the spectral intensity (cf. labels in inset to Figure 6a). The two main bands of staphyloxanthin, namely, the C–C and C=C stretching bands from the carotenoid chain, were obviously missing in bone samples inoculated with only PBS and in the non-inoculated healthy ones. Conceivably, their relative intensity increased with increasing concentration of inoculated *S. aureus* bacteria, as shown in the plot of the areal ratio, *R_inf_
*= *I*_1172_/*I*_961_, as a function of initial bacterial concentration given in Figure 6b. The spectroscopic ratio *R_inf_* could thus be assumed as an overall index of infection for osteomyelitic bones. An interesting finding was that C–C and C=C staphyloxanthin chain signals both tended to systematically shift towards lower wavenumbers with increasing bacterial concentration. This point will be discussed in detail in the forthcoming Section 3.2. Regarding cytochrome chemistry, we specifically monitored two additional spectroscopic parameters, namely, the areal ratios, *R_imm_
*= *I*_1596_/*I*_961_ and *R_cyt_
*= *I*_1644_/*I*_750_. The spectral ratio, *R_imm_*, which gives the enhancement in cytochrome *c* signal normalized to the bone apatite signal, could be assumed as an index for the level of immune response in the bone tissue in response to the presence of bacteria. On the other hand, the ratio, *R_cyt_*, which gives the relative intensity of the oxidized cytochrome *c* marker to the overall cytochrome *c* marker, could be assumed as an index for cytochrome *c* oxidation state. As shown in Figure 6c, both these parameters increased with increasing bacterial concentration. For the sake of completeness, we should also emphasize that cytochrome *c* signals were conspicuously missing in spectra collected at the surface of healthy bone (cf. Figure 3a) because (i) in a healthy cellular environment, the amount of cytochrome *c* is enzymatically reduced; and (ii) cortical bone (i.e., the only tissue probed when the Raman probe is focused on the bone surface) is mostly made up of mineralized extracellular matrix and contains relatively few cells (mainly osteocytes) as compared to bone marrow, which is instead highly cellular. Finally, note that the C=O signals (located within the interval 1700~1750 cm^−1^; cf. labels in inset) also increased with increasing bacterial concentration (cf. arrows in Figure 6a). While this spectroscopic feature represents a clear sign for enhancement of the overall oxidation state in bone marrow environment, it was not possible at the present stage to locate the specific molecules to which such signals were actually related.

Figure 7a shows a rat tibia initially inoculated with 100 μL PBS containing a bacterial concentration of 10^8^ CFU/mmL and then extracted from the rat body at 7 d post inoculation. Bacterial injection was made in correspondence with the location labeled as point 1, while Raman measurements were made at points 1 to 4, i.e., increasingly distant from the inoculation site and approximately spanning 1 cm from each other (cf. insets in Figure 7a). Average spectra collected at points 1~4 are displayed from top to bottom in Figure 7b.

In the spectra collected at points 1 to 4, we first located the marker signals for hydroxyapatite, staphyloxanthin, and cytochrome *c* (cf. labels and wavenumbers in inset). Then, we computed the ratios *R_inf_* and *R_imm_* as indexes of bacterial concentration and immune response in bone tissue, respectively. As shown in Figure 7c, both spectroscopic parameters tended to decrease with increasing distance from the inoculation point 1. This result is consistent with the view of an acute and yet expanding infection. This could be considered as further evidence that Raman spectroscopy is capable of non-destructively giving quantitative information about osteomyelitis progress inside an infected bone.

### 2.4. Raman Spectra of Bone at the Subacute Stage of Osteomyelitis

Figure 8a shows a comparison between average Raman spectra collected on rat tibias inoculated with 100 μL PBS–10^8^ CFU/mmL bacterial concentration and then extracted from the rat body after 7 and 35 d post inoculation (upper and lower spectrum, respectively; cf. labels in inset). The infection at 35 d could be considered as entering in its subacute stage of osteomyelitic pathology [43].

As seen, the morphologies of the average spectra collected at 7 and 35 d appeared quite different to each other, although the main markers of hydroxyapatite, staphyloxanthin, and cytochrome *c* (both reduced and oxidized) could yet be clearly resolved in both cases. We computed the three spectroscopic parameters so far established, namely, *R_inf_*, *R_imm_*, and *R_cyt_*. The results of these computations are compared in Figure 8b for samples in acute (7 d) and subacute (35 d) stages. As seen at 35 d post injection, namely, in the subacute stage, both levels of infection and immune response were significantly reduced, while the degree of cytochrome oxidation showed a more than twofold increase. Additional features that differentiated the spectrum collected at 35 d from the one at 7 d were analyzed in search for a common origin (cf. labels in inset to Figure 8b, lower spectrum). A search in the Raman literature for possible biomolecules located a clear matching with the spectrum of sphingomyelin (cf. spectrum of the elementary molecule in Figure 8c) [44,45]. This finding relates the lipidomics of bone marrow to the presence of abscesses, and it is a consequence of the accumulation of inflammatory cells, such as neutrophils and other leukocytes, of which populations significantly increase at infection sites. Similarly, spectral features also suggest an abundance of other related lipid structures, such as ceramide and sphingosine (also shown in Figure 8c; cf. labels in inset) [46,47], both also present in the neutrophil population. The spectrum at the infection site 35 d post infection showed enhanced signals from choline group N^+^(CH_3_)_3_ symmetric stretching (at ~700 cm^−1^), CH_3_ rocking (at 888 and 900 cm^−1^), P–O stretching (at 1055 cm^−1^), asymmetric and symmetric C–C stretching (at 1063 and 1130 cm^−1^, respectively), CH_2_ twisting and scissoring (at ~1283 and 1480 cm^−1^, respectively), and C=O stretching (at 1700 and 1735 cm^−1^) (cf. labels in inset to Figure 8a–c) [44]. In the metabolic cycle of neutrophils, fractions of sphingomyelin, ceramide, and sphingosine are regulated by the activities of two enzymes, namely, sphingomyelin synthase and ceramide kinase (cf. schematic draft in Figure 8d). Both enzymes are crucial for neutrophil phagocytic activity. This point will be discussed in detail in the forthcoming Section 3.1. Finally, it should be noted that Raman peaks of carotenoids at ~1522 cm^−1^ is best excited with 532 nm due to resonance enhancement, making it a very strong signal. Meanwhile, fluorescence emission from porphyrins and bone-collagen autofluorescence are also expected in the wavenumber interval 1300~1500 cm^−1^ [48]. Besides the lower wavenumber of the above fluorescence signals, both porphyrin fluorescence and bone autofluorescence are broad and much weaker than the resonance-enhanced carotenoid band [48]. In other words, the strong resonance Raman signal at 1522 cm^−1^ definitely outcompetes background fluorescence, even if some tail-overlap might exist. Such overlap was thus fully eliminated through background subtraction.

## 3. Discussion

### 3.1. Osteomyelitis and Its Link to Bone Immune Response

Osteomyelitis is a debilitating infectious disease that involves painful bone inflammation and infection; it can, in principle, affect any bone in the body, but preferentially occurs in long bones, typically developing in distal femur and proximal tibia [49]. Osteomyelitis infections could reach a bone by haematogeneous routes, or could spread from contiguous sources secondary to bone-open injuries, caused by trauma, surgery, or joint replacement [50]. The present model study features this latter type of infection. Independent of source and spreading path, osteomyelitis infection generally propagates inside the bone through the formation of independent infection loci. Such infected islands detach from the healthy bone tissue surrounded by a ring of sclerosis [51]. When osteomyelitis reaches an acute or chronic stage, uncontrolled bone remodeling takes place and bone-associated deformity is observed [52]. Among osteomyelitis cases, *S. aureus* is a common culprit, this bacterium being capable of adhering and becoming internalized in a variety of host cells including osteoblasts [50,52]. As the leading cause of osteomyelitis, *S. aureus* possesses several immunoevasive strategies, which include its ability of becoming surrounded by immune cells and impeding neutrophils clearance, thus gaining protection from both host response and antibiotic treatments [53,54,55,56].

The present Raman investigation builds upon previously proved notions of *S. aureus* pathology and well-established medical practice to diagnose bone osteomyelitis. We propose here a real-time procedure applicable in the surgery room and based on quantitative spectroscopic parameters, which could be used as biomarkers to judge non-destructively the extent of bone internal infection and the degree of apoptotic response of bone tissue. It is shown that by merely shining a laser beam of visible wavelength on the infected bone, one could concurrently estimate both bacterial concentration and bone immune response. Three Raman spectroscopic parameters were proposed as follows: (i) the relative intensity ratio, *R_inf_
*= *I*_1176_/*I*_961_, which links exponentially to pathogen density/multiplication and could thus be referred to as the “*infection index*”; (ii) the relative intensity ratio, *R_imm_
*= *I*_1596_/*I*_961_, which gives the level of immune response in bone tissue and could thus be referred to as the “*immune response index*”; and (iii) the relative intensity ratio, *R_cyt_
*= *I*_1644_/*I*_750_, which could be defined as the “*cytochrome chemistry index*″, since it shows the level of oxidation in mitochondria cytochrome *c* (i.e., the loss of electrons in its haem ring; cf. Figure 5) in the eukaryotic cells of the infected bone tissue; since oxidation occurs when apoptotic cells release cytochrome *c* in the cytosol, the *R_cyt_* parameter could also be referred to as the “*apoptosis index*”.

After one-week exposure to *S. aureus* inoculation, the infection index, *R_inf_* (cf. above point (i)), of the rat tibia was found to exponentially link to the initial bacterial concentration. While the areal ratio of bacteria vs. bone tissue signals at the time of inoculation obviously only depends on bacterial dose, a strongly non-linear dependence on the initial concentration of inoculated bacteria could be observed after one week (cf. Figure 6b). This observation can be explained by noticing that the bacterial population in the case of bone infections by *S. aureus* usually increases exponentially during the acute stage (i.e., up to 7 d) provided that essential nutrients are available (i.e., which is the case here for infected bone marrow) [57]. The increase in immune response index, *R_imm_* (cf. above item (ii), which features a signal from the haem ring of cytochrome *c*, also followed an exponential trend as a function of initial bacterial concentration (cf. Figure 6c). This latter trend is also conceivable upon considering that the degree of immune response should follow the spread of bacterial infection. Cytochrome *c* is a mitochondrial molecule that reflects danger-associated molecular patterns in circumstances leading to cell injury [58]; its release into the extracellular space in pathological conditions is a marker of severe mitochondrial damage and cell death. Under pathological circumstances, cytochrome *c* molecules are recognized by specific pattern recognition receptors of the immune cells leading to inflammation. Accordingly, the extent of cytochrome *c* formation, which can easily be measured by Raman spectroscopy, may become a clinically useful biomarker for both diagnosing the level of inflammation and assessing the severity of osteomyelitis and its related pathological circumstances. The Raman detection of high cytochrome *c* levels into the bone marrow reflects its release into the extracellular space. In support of the above interpretation, the presence of a number of other biomarker molecules for cellular damage, including mitochondrial DNA, has been discussed in the published literature [59,60,61]. In other words, the level of cytochrome *c* circulating within the bone marrow, namely, the *R_imm_* ratio non-destructively detected by Raman spectroscopy, could be assumed as an indirect but reliable marker for osteomyelitis.

When cytochrome *c* is released from mitochondria into the cytosol, formation of the apoptosome is triggered, which results in activation of caspases [62,63]. Hancock et al. [63] were the first to propose that the redox state of cytochrome *c* regulates apoptosis, and this hypothesis was later supported by further studies by Pan et al. [64] and Suto et al. [65] In healthy cells, enzymes, and reductants can rapidly reduce cytosolic cytochrome *c*, thus blocking apoptosis. Conversely, cytosolic cytochrome *c* undergoes rapid oxidation in apoptotic cells by mitochondrial cytochrome oxidase while accessing it through permeabilization of the outer membrane [66]. Although a complete explanation of the exact mechanisms triggering caspase activation is yet to be given, it is now proved that only the oxidized form of cytochrome *c*, namely, the presence of Fe^3+^ in the haem ring, can induce caspase activation via apoptosome [67]. Accordingly, monitoring the regulation of the redox state of cytochrome *c* potentially enables one to assess the extent to which the intrinsic pathway of apoptosis has developed, at least at a relatively late stage. The present study builds upon the above understanding and exploits the unique circumstance of high Raman cross section for the haem rings of cytochrome *c* coupled with their signal diversity in reduced (Fe^2+^) and oxidized (Fe^3+^) states [36,37,38,39,40,68]. Since the spectroscopic ratio, *R_cyt_
*(i.e., the above item (iii)), directly measures the volume fraction of oxidized vs. reduced haem rings, its Raman assessment provides an additional straightforward path to the physiological state of bone cells; the higher the ratio, the higher the number of apoptotic osteoblasts. In a recent paper, Abramczyk et al. [69] used UV-VIS electronic absorption and Raman spectroscopies to unfold the molecular details of the iron–oxygen interaction and metal spin state in haem proteins. Their study experimentally confirmed the hypothesis that apoptosis is controlled by the axial ligands and the redox status of the iron ion oscillating between Fe^2+^ and Fe^3+^ in the haem group. This finding in turn supports our hypothesis that the Raman ratio, *R_cyt_*, could be a measure of the fraction of apoptotic osteoblasts. In a recent review, Villalpando-Rodriguez and Gibson [70] have examined the effect of the level and type of reactive oxygen species in regulating cell death as signaled by the specific radicals involved. The increase in reactive oxygen radicals upon cell death arises from different sources within a cell, which include damage to cellular organelles and plasma membrane. Such radical production adds to the radicals created by neutrophils in their bactericidal role [71]. Marriott [72] studied apoptosis mechanisms specifically for osteomielytic bone. Besides the presence of extracellular bacterial components and bacterial internalization, which greatly reduce osteoblast viability and maximize osteoblast apoptosis, the causative agents of osteomyelitis are also capable of inducing continuous osteoblast apoptosis upon activating intrinsic and extrinsic cell death pathways that uncouple bone formation and resorption. Summarizing data of seven-day infections, we propose here that the three spectroscopic parameters *R_inf_*, *R_imm_*, and *R_cyt_* relate to each other and give us a comprehensive and concurrent view of level of infection, degree of immune response of bone tissue, and stage of osteoblast apoptosis.

Raman characterizations after 35 days since bacterial inoculation gave us confirmation of 7-day assessments and provided further insight into osteomyelitis development. As shown in Figure 8a, the average spectrum of 10^8^ CFU/mmL-inoculated tibias at 35 d, although conserving all the spectral characteristics related to both eukaryotic and prokaryotic cells discussed in the analysis of 7-day tibias, also showed important variations in its morphology. Analysis of the three spectroscopic parameters *R_inf_*, *R_imm_*, and *R_cyt_* (cf. Figure 8b) revealed that *infection index* and *immune response index* both decreased by about one third as compared to spectra collected after 7-days exposure. On the other hand, the *cytochrome chemistry index* doubled as a sign of a larger presence of apoptotic osteoblasts. These data can be likened to the scenario of a battlefield when the conflict is going to terminate, with less eukaryotic and prokaryotic cells fighting each other and many eukaryotic cells sacrificed in the battle. General notions on how osteomyelitis develops support the idea of such a scenario. On day 7, we observed a stage of acute osteomyelitis, which generally lasts about two weeks after disease onset, while at day 35 the infection reached the stage of subacute osteomyelitis, which generally lasts up to several months before reaching the chronic stage [23,73]. According to Masters et al. [74], invasion and survival of *S. aureus* in bone tissue is strongly enhanced in comparison with other bacteria due to the capacity of this bacterium to form robust abscess communities. Abscess communities encase themselves within a protective barrier in order to be protected from blood flow. In this way, they become capable of persisting for prolonged periods of time and gain protection from neutrophil clearance [28,55,75]. Moreover, such communities exploit the host response for their own protection [76,77]. The protective activity starts with the formation of a protective fibrous pseudocapsule, which becomes successively surrounded by a large number of necrotic immune cells, mostly neutrophils, as a result of direct and indirect killing [78,79]. Since the bacterial cells that live in abscess communities are largely devoid of nutrient and oxygen, they tailor gene expression according to nutrient availability. In particular, they manage to scavenge iron by means of iron-scavenging proteins that bind hemoglobin and extract haem from eukaryotic cells’ mitochondria as a precious source of iron (and oxygen, since haem released in the cytosol are oxidized; cf. Figure 4) [80,81]. In other words, the trend variation for *R_cyt_* at 7 and 35 d, as reported in Figure 8b, which shows a strong enhancement in oxidized haem, should link to an active “squeezing” of haem out of surrounding cells and, thus, to a scenario of a widely spread development of abscess communities within bone marrow.

An additional feature in the spectra of Figure 8a consisted in the appearance of strongly enhanced Raman signals in the average spectrum at 35 d as compared to at 7 d. A total of seven such signals could be located (cf. arrows and wavenumbers in inset to the lower spectrum of Figure 8a). As previously described, all these additional bands could be traced back to increased fractions of ceramide and sphingolipids at infected sites. Good et al. [82] investigated the lipidomics of bone marrow abscesses by means of imaging mass spectrometry and microscopy. The lipidome of *S. aureus*-infected murine femurs were found to link to the metabolic and signaling consequences of infection. A spatially resolved map located about 250 different lipid molecules at healthy and infected sites throughout the length of murine femurs, with different composition profiles being found for different types of tissue. In bone marrow abscesses, significant structural alterations were revealed. At abscess sites, an increased presence of ether lipids could be detected independent of the infection stage. This circumstance could be traced back to the presence of inflammatory cells, since ether lipids are abundant in neutrophils and other leukocytes [83,84,85,86,87]. When bacteria invade the intramedullary cavity, the population of myeloid lineage cells significantly increases, a process usually referred to as myelopoiesis [88]. Concurrently, increases in ether lipids occur in stem cells due to both membrane remodeling and lipid-mediated signaling purposes. It follows that an increased presence of ether molecules is a mark for abscess formation and related inflammatory signaling. Accordingly, Raman signals from ether lipids might be considered as biomarkers for neutrophils and other myeloid lineage cells present in the abscess. However, this assertion will need additional experimental evidence to be confirmed. Additional classes of lipids copiously produced in the inner portion of abscesses and throughout the intramedullary cavity of *S. aureus*-infected bones are sphingolipids and acyl-glycerophospholipids [82]. Abundance of such lipids in the neutrophil population within the abscess is an unequivocal index for sphingomyelin synthase and ceramide kinase activities. Both enzymes are crucial for neutrophil phagocytic activity [89,90]. As confirmed in the present Raman investigation, enhanced lipid synthesis hints at abscess-located neutrophils undergoing apoptosis. As shown by comparing the spectrum of infected bone at 35 d with the reference spectra of elementary sphingolipids (cf. Figure 8a and 8c, respectively), the observed enhancement in peculiar Raman signals might represent a boost in sphingomyelin synthase and ceramide kinase activities in leukocytes (cf. Figure 8d) [91]. Isogai et al. [92] used bioluminescence imaging and plasma metabolome analyses in a mouse osteomyelitis model similar to the one adopted in the present study. Consistent with our results, those researchers identified sphingosine as the highest loading factor. As schematically shown in Figure 8d, sphingosine is liberated from cell membrane sphingomyelin through a series of reactions catalyzed by metabolic enzymes, including sphingomyelinase and ceramidase [93,94,95]. Under physiological conditions the sphingomyelinase enzyme is stably located within lysosome organelles. However, under stress, the organelles release the enzyme, which undergoes acidification. Acid sphingomyelinase then hydrolyzes the sphingomyelin located at the outer leaflet of membranes and transforms it into ceramide, which in turn is deacetylated into sphingosine through ceramidase that cleaves out the fatty acid moiety from the ceramide structure (cf. Figure 8d) [96,97]. According to Cheung and Claus [98], *S. aureus* stimulates acid sphingomyelinase through the release of reactive oxygen species, which results in ceramide release. Accumulation of reactive oxygen species is the consequence of an altered redox status that follows from acid sphingomyelinase activation and ceramide generation, a common event in neutrophil apoptosis. Neutrophil death by apoptosis is thus central to hemostasis and to the resolution of inflammation [99,100,101].

In summary, the present Raman data, in agreement with previous studies of osteomyelitis metabolome, provided molecular insight into the metabolism of the eukaryotic cells present at the fibrotic border of abscesses through the enhancement of specific spectroscopic biomarkers that can discern the immune activity of neutrophils, leukocyte, and macrophages.

### 3.2. Bacteria Structural Modification in Response to Host Immune Response

After becoming exposed to cellular-rich marrow within the intramedullary cavity of long bones, *S. aureus* exploits a broad range of virulence factors in order to colonize, proliferate, and expand towards farther regions [53,102]. Bacteria first affix to cells and bone extracellular matrix, then they start recruiting innate leukocytes and deliver toxins and immunoevasive factors to escape immune defenses. Within several days, the resulting interaction between host and pathogen develops into the formation of an inflammatory zone with the distinctive architecture of an abscess. The abscess structure generally consists of a central bacterial microcolony, commonly referred to as staphylococcal abscess community, surrounded by layers of necrotic and viable neutrophils and leukocytes positioned around a fibrous layer [53]. Such a biotic assembly effectively sequesters the necrotic abscess from healthier surrounding tissue and serves to establish a structural barrier between viable leukocyte defenses and the *S. aureus* proliferating colony, thus impairing vascular paths and discontinuing antibiotic delivery and treatment [103]. An additionally important tool in the overall immune-escape strategy is the intrinsic *S. aureus* capacity of neutralizing oxygen radicals from viable neutrophils. The present study supports the hypothesis that the presence of carotenoid staphyloxanthin in the *S. aureus* structure facilitates the building up of the necrotic layer and busts up resistance against viable neutrophils. The presence of staphyloxanthin as a cell membrane pigment in the *S. aureus* structure is a long-known characteristic [104]. The variable production of this carotenoid molecule has been reported to be a strain-dependent characteristic that modulates the intrinsic oxidative stress resistance of the bacterium [105]. In highly oxidative environments, *S. aureus* succeeds in preventing and mitigating damages to its protein and genetic molecules by exploiting carotenoid staphyloxanthin as a neutralizing agent of the reactive oxygen species produced by neutrophils and apoptotic osteoblasts (cf. Section 3.1) [106,107].

In this study, in situ epidemiologic insights could be obtained from Raman spectra of infected bone upon analyzing the metabolic state of the *S. aureus* prokaryotic cells through its characteristic staphyloxanthin signals. For this purpose, we monitored the degree of oxidation of staphyloxanthin through examining its characteristic C–C and C=C Raman signals nominally located at 1172 and 1522 cm^−1^, respectively (cf. reference staphyloxanthin structure and Raman spectrum in Figure 5a and 5b, respectively). Figure 9a shows a plot of areal ratio *I*_1172_/*I*_1596_ as a function of initial bacterial concentration inoculated in bone marrow. The two selected Raman markers represent the C–C bond in the carotenoid staphyloxanthin chain (at ~1172 cm^−1^; cf. Figure 4b) and the main signal of cytochrome *c* in osteoblasts (at ~1596 cm^−1^; cf. Figure 5b). Note that the spectral location of the former signal was indicatively given as 1172 cm^−1^, as in the case of the stoichiometric reference molecule in Figure 5. However, at the infection sites, the spectral location of this band systematically shifted toward higher wavenumbers with respect to the reference molecule. This point will be discussed later. The plot in Figure 9a shows a linear increase in the spectral ratio *I*_1172_/*I*_1596_ with increasing bacterial concentration, thus confirming that the observed level of osteoblast apoptosis, represented by the fraction of cytochrome *c* released in the cytosol, actually links to the presence of bacteria, as previously discussed in Section 4.2. On the other hand, Figure 9b features the Raman areal ratio *I*_1522_/*I*_1172_ between the relative intensity of C=C and C–C stretching signals. This ratio is directly related to the number of double vs. single carbon/carbon bonds in the carotenoid chain or, in other words, the carotenoid chain length, whereby the shorter the chain, the lower the ratio [108,109,110]. Note again that the wavenumber of the C=C stretching signal refers to the spectrum of the reference molecule in Figure 5b. The plot in Figure 9b revealed that bacteria belonging to infected bones initially inoculated with lower doses contained staphyloxanthin molecules with shorter carotenoid chain lengths. The plot also shows that the *S. aureus* culture, namely, bacteria in stress-free physiological equilibrium, experienced the highest ratio among the measured spectra of bacteria (cf. labels in inset). The cultured *S. aureus* strain, which could proliferate under stress-free conditions, included staphyloxanthin molecules with a chain only ~15% shorter than that of stoichiometric molecules (cf. Figure 9b). A confirmation for the trend in Figure 9b could be obtained by examining the wavenumber at maximum for both C–C and C=C stretching signals, whose shifts towards higher values were reported to be markers for shortened carotenoid chain lengths [110]. We consistently found that staphyloxanthin molecules in stress-free *S. aureus* culture experienced the closest C–C and C=C stretching wavenumbers to those of stoichiometric staphyloxanthin (namely, 1176 vs. 1172 cm^−1^ and 1535 vs. 1522 cm^−1^, respectively). On the other hand, Raman measurement performed on bone samples with different initial concentrations of bacteria tended to display C–C and C=C stretching signals at higher wavenumbers; the lower the initial bacterial dose the higher the wavenumber of the C–C stretching signal. Figure 9c shows both C–C and C=C stretching wavenumbers as detected for different (initial) bacterial doses (cf. labels in inset) plotted against the average number, *N*, of C=C bonds present in the chain of staphyloxanthin molecules, according to calibrations reported by Schaffer et al. [108] Both plots follow a linear trend of increasingly lower number of C=C bonds with decreasing initial dose of inoculated bacteria from the lowest wavenumbers of 1172 and 1522 cm^−1^ (corresponding to *N* = 10 in stoichiometric molecules) to ~1187 and ~1561 cm^−1^ (corresponding to *N* ≅ 6.5, as directly deduced from the plot in Figure 9c) for 10^3^ CFU/mmL initial inoculation dose. Note, however, that while the molecular origin of the observed shift unequivocally relates to chain length, the quantitative calibrations shown in Ref. [107] do not necessarily match the present data in a quantitative way, mainly because of differences in pH between experiments in vivo and in solution. According to Krinsky and Yeum [111], in the presence of oxygen radicals, interruption of the conjugated carbon/carbon double bonds can occur upon central cleavage or excentric cleavage. In the former process, an apocarotenoid molecule is formed, while the latter reaction results in the formation of a series of apocarotenals [112] and epoxides [113] that can further be broken down into β-apocarotenoic acids [114]. The above molecular splitting process can occur because, in the reaction between a carotenoid carbon chain and oxygen radicals, the conjugated double bonds can be easily oxidized, losing an electron from the polyene chain to form a radical cation [115,116]. Reactive oxygen species include superoxide anion radicals (O^•−^), hydroxyl radical (^•^OH), and hydrogen peroxide H_2_O_2_ endogenously produced by mitochondria of apoptotic osteoblasts and neutrophils during infection. In acidic environment, cleaved chains are expected to end with carbonyl bonds [111]. Based on Ref. [111], Figure 9d shows a schematic draft for the possible mechanisms of interaction between the carotenoid chain of staphyloxanthin and free oxygen radicals produced by apoptotic osteoblasts and neutrophils. Note that the present data do not allow discerning the relative fraction of apocarotenoid/apocarotenals molecules generated upon staphyloxanthin chain cleavage, but only an average number of C=C bonds per molecule can be computed. An examination of signal enhancement in the carbonyl zone (1700~1750 cm^−1^) suggests that carbonyl terminals start to significantly appear at higher bacterial concentrations (cf. spectrum at 10^8^ CFU/mmL and arrows/labels in inset to Figure 6a). The Raman spectroscopic ratio, *I*_1522_/*I*_1172_, consistently with C–C and C=C stretching wavenumber shifts, clearly points to longer carotenoid chains in samples with a higher initial dose of inoculated bacteria. These experimental evidences could be explained by considering the peculiar kinetics of *S. aureus* bone infections (cf. Figure 10). The immune response of bone tissue, and thus the amount oxygen radicals produced by both apoptotic osteoblasts and neutrophils, will obviously tend to increase with increasing bacterial concentration (cf. Figure 9a). However, according to the model of central bacterial microcolony/staphylococcal abscess community proposed by Cheng et al. [53], if the surrounding necrotic layer tends to become thicker, shielding could become more effective as the bacterial population increases. Accordingly, bacteria from sites with denser bacterial enclaves receive less damage from oxygen radicals and thus display, on average, longer carotenoid chain lengths. Raman data collected at 35 d after inoculation somehow confirmed the above model. The computed spectroscopic ratios *I*_1172_/*I*_1596_ and *I*_1522_/*I*_1172_ long-term after inoculation significantly decreased by about 50% of their respective values after 7 d exposure (cf. data and labels in inset to Figure 9a and 9b, respectively). According to the above-described biological meaning of these two ratios, it follows that a decrease in the spectral ratio *I*_1172_/*I*_1596_ actually shows that the level of osteoblast apoptosis, namely, the fraction of cytochrome *c* released in the cytosol, is significantly higher at 35 d than at 7 d after inoculation. On the other hand, a lower value of the ratio *I*_1522_/*I*_1172_ indicates an average of shorter length carotenoid chains and thus a higher amount in “sacrificial” cleavages of C=C bonds in response to oxygen radical attack. Note that the concurrent decrease in both spectroscopic ratios is a consistent result since an increase in apoptotic cells involves a higher production of reactive oxygen radicals from damaged cellular organelles and plasma membrane. This in turn produces a higher fraction of cleaved C=C bonds in the carotenoid chain in order to neutralize the radicals. Remarkably, analyses of wavenumber shift were consistent with relative intensity data (cf. data and labels in inset to Figure 8a and Figure 9c).

### 3.3. Raman Insights into S. aureus Persistence and Immunity Manipulation

Among a plethora of powerful strategies to counteract immune system, *S. aureus* is capable of secreting an assortment of immune evasion toxins that can cleave key molecules in the host cells and disrupt the integrity of extracellular matrix [76,77]. Proteins secreted by *S. aureus* can inhibit the antibacterial function of neutrophils and lymphocytes, while concurrently manipulating apoptotic, necroptotic, and pyroptotic modes of cell death, thus potentiating survival chances and supporting the establishment of the infection. An important aspect of the invasive strategy of *S. aureus* is its capacity to induce host cells to produce highly diffusible molecules that cause the death of the hematopoietic stem cells present in bone marrow. Since such immature cells are progenitors of neutrophils, white blood cells, red blood cells, and platelets, the pathways to hematopoietic cell death enables *S. aureus* to comprehensively shape an immune strategy conducive to persistence. Such strategy juxtaposes with mechanisms of immune evasion, autophagy escape, and tolerance to intracellular killing as commonly exploited by other invasive bacteria. An additional path to persistence in abscess formation consists of *S. aureus* secreting proapoptotic molecules targeting phagocytes and signaling molecules that activate apoptosis of immune cells; this de facto excludes macrophages from abscess lesions without causing inflammation [117,118,119]. In other words, *S. aureus* possesses a unique skill in exploiting apoptosis to incapacitate macrophages and other host cells without provoking inflammatory responses.

The pathological output of *S. aureus* invasion in bone marrow is the formation of abscess lesions. A notable feature of staphylococcal abscesses resides in their massive infiltration of immune cells in response to cellular destruction and proinflammatory signals. Early in post infection, staphylococcal abscess formation proceeds in a disordered manner with tissue infiltration of a large numbers of immune cells, predominantly neutrophils and macrophages; within five days, however, staphylococcal abscess communities converge at the center of the lesion and become enclosed within fibrin deposits with surrounding concentric layers of immune cells. At this stage, apoptotic neutrophils appear arranged in concentric layers together with other immune cells, including macrophages and lymphocytes (Figure 10a) [80,117]. Beyond two weeks post infection, most immune cells become necrotic and cellular detritus accumulate, with macrophages being positioned at the periphery and, together with lymphocytes (as foamy cells), slowly being pushed toward farther regions (Figure 10b) until eventual rupture of the abscess and release of purulent exudate/staphylococci for dissemination into new abscesses.

In a recent paper, Kobayashi et al. [78] reviewed the current knowledge of mechanisms and processes behind the formation of *S. aureus* abscesses, including the fundamental involvement of neutrophils. The proposed scenario is comprehensive of three principal activities: recruiting neutrophils, making them contribute to causing host cell lysis, and building up the fibrin capsule that surrounds the abscess. The present Raman experiments provided new molecular insight into the above pathogenic notions. In particular, we were able to locate specific Raman markers that allowed the quantitatively linking of host cell lysis to the level of bacterial infection. Moreover, examination of staphyloxanthin markers from bacteria at the early stage of infection clearly revealed the strategy that *S. aureus* adopts to resist the shower of oxygen radicals that it induced and contributed to mount up, sacrificing double bonds in the carbon chains of its staphyloxanthin molecules. At a later post infection stage, the scenario conspicuously changes, reaching a stable “post-war” situation, namely, a high level of apoptotic eukaryotic cells left on the “battlefield” and surviving bacteria with shortened staphyloxanthin chains. Figure 10, which is based on the schematic draft proposed in Ref. [120], attempts to summarize the Raman findings in the pathogenic context induced by *S. aureus* abscesses. As discussed, abscess development and related chemical reactions differed in (a) acute (7 days) and (b) subacute (35 days) stages of infection. Oxygen radicals induce apoptosis in eukaryotic cells, while bacteria enclosed in the central microcolony of the abscess survive upon sacrificial cleavage reaction of their staphyloxanthin molecules.

**Figure 10 ijms-26-08572-f010:**
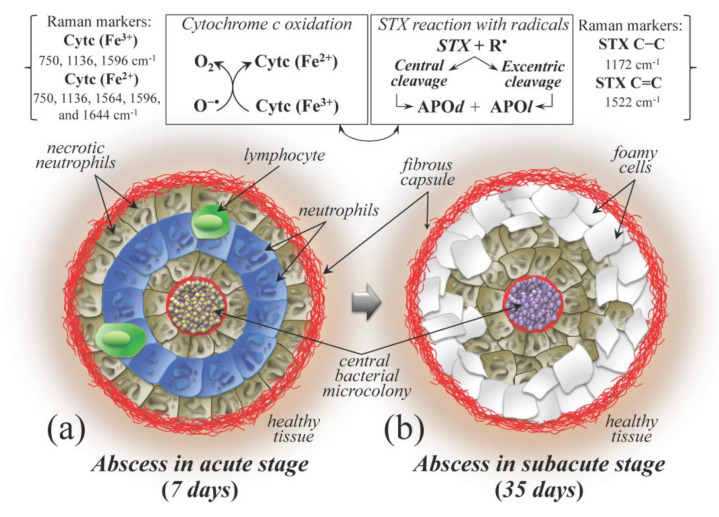
Schematic draft [120] summarizing the *S. aureus* abscess development with respect to both its pathogenic context, the related chemical reactions, and Raman markers (cf. square insets and labels on top), as revealed by Raman spectroscopic assessments, at (**a**) acute (7 days) and (**b**) subacute (35 days) stages. Abbreviations: R^•^ → radicals, APO*d* → apocarotenoid, APO*l* → apocarotenal. Other abbreviations are the same as in Figure 6, Figure 7 and Figure 8.

### 3.4. The Potential Benefits of a Raman Probe in the Surgery Room

As mentioned in the introduction, in assessing osteomyelitis, nowadays surgeons mainly rely on preoperative imaging techniques such as X-ray radiography, computed tomography scans, and magnetic resonance imaging. However, these methods can merely provide morphological information, while lacking direct biochemical insights. Accordingly, a complete picture including molecular-scale information is hardly available, especially in cases of complex anatomy. During surgery, inflammation associated with osteomyelitis generally alters the appearance and texture of tissues, making it challenging to visually differentiate between healthy and infected areas. Precisely locating sites of bone necrosis could also be challenging. To counteract the above hindrances, advances in imaging technology, intraoperative monitoring, and surgical techniques have been proposed in order to improve accuracy in assessing osteomyelitis during surgery [5]. In this latter context, we show here that a potential contribution of the Raman probe in the surgery room could represent a breakthrough in supporting intraoperative decisions. The present study clearly shows that promptly retrievable Raman parameters could enhance the precision of osteomyelitis assessment during surgical procedures. In addition to the possibility of using Raman spectroscopy to analyze in situ molecular alterations in bone tissue itself, as proposed by other authors [10,16], we show here how non-destructively retrieved spectroscopic parameters could be directly related to the level of infection inside the bone as well as parameters linked to metabolism and physiology of both eukaryotic and prokaryotic cells coexisting in osteomyelitic bone tissue. Changes in vibrational signals from specific fingerprint molecules, such as cytochrome *c* and staphyloxanthin in cells and bacteria, respectively, could be indicative of the level of infection and inflammation, and of the extent of cell apoptosis and bone necrosis. This in turn could enable surgeons to correctly assess in real time if there is a need to remove additional portions of the bone during surgery.

### 3.5. Limitations of This Study

While the non-invasive nature of the Raman technique guarantees minimally disruptive assessments and is promising in surgical setting, clinical applications of Raman spectroscopy to assess osteomyelitic patients are still in their infancy. Challenges will include development of spectrally resolved fiber Raman devices, refinement and standardization of Raman algorithms, and final validations in larger patient cohorts. Investigation will also be needed on how factors like tissue heterogeneity, variations in bone density, or previous treatments may affect the spectral signatures. Given that human cortical bone, particularly in the tibia, is denser and thicker than in the present Wistar rat animal model, new strategies or modifications of laser power/wavelength are necessary to adapt and implement the Raman spectroscopy method to be effective in human clinical studies.

## 4. Methods and Materials

### 4.1. Animal Experiments and Sample Preparation

The present experiments used 5~6 weeks old male Wistar/ST rats. *Staphylococcus aureus* (*S. aureus*; strain ATCC6538) was purchased from the Biological Resource Center, National Institute of Technology and Evaluation (NITE NBRC; Tokyo, Japan). Rats were kept in a 12 h light/dark cycle with free access to food and water. In cases where deterioration of general condition, eating disorders, or sudden weight loss of 20% or more was observed, euthanasia was performed by overdose of barbiturates as a humane endpoint. Rats were anesthetized by administering intraperitoneally a mixture of 0.375 mg/kg medetomidine, 2.0 mg/kg midazolam, and 2.5 mg/kg butorphanol. A skin incision was made just above the knee joint of the hind limb of each rat. The patellar tendon was cut longitudinally, and a bony hole was drilled at the tibial plateau with 1.0 mm diameter wire. Suspensions of *S. aureus* in phosphate-buffered saline (PBS) were injected into the medullary cavity of one of the tibias in the neighborhood of the medial condyle in doses 10^3^~10^8^ CFU/mmL in order to create an osteomyelitis model. For comparison, tibia samples injected with the same amount of PBS without bacterial contamination were also produced and followed up with the exact same procedure of the contaminated ones. One or five weeks after injection, rats were euthanatized and both their healthy tibias and tibias affected by suppurative osteomyelitis were removed. In this way, two samples could be obtained for each sacrificed animal. Then, the spectra of healthy and infected bone tissues were analyzed using Raman spectroscopy. Observations of *S. aureus* bacterial colonies in the tibia medullary cavity were made by means of a fluorescence microscope (BZX710; Keyence, Osaka, Japan).

### 4.2. Raman Spectroscopic Assessments and Spectroscopic Data Treatments

Reference Raman spectra were first collected in situ on as-cultured *S. auris* bacterial samples and healthy rat tibia; then, spectra were collected on tibia samples injected either with bacteria-contaminated PBS or with uncontaminated PBS, according to the description given in Section 4.1. Raman data were collected by means of a spectrometer designed for measurements on biological samples (LabRAM HR800, Horiba/Jobin-Yvon, Kyoto, Japan). The equipment launched an optical circuit (with a 20× objective lens) employing a holographic notch filter, which concurrently enabled high signal efficiency and suitable spectral resolution. The wavelength of the incoming light was 532 nm, as generated by a solid-state laser source operating at 10 mW. Note that the selected wavelength for the incoming laser is expected to represent a resonance scattering wavelength, thus producing sharp and intense signals from cytochrome *c*. The Raman scattered light was monitored by means of a single monochromator interfaced with an air-cooled charge-coupled device (CCD) detector (Andor DV420-OE322; 1024 × 256 pixels; Belfast, UK). The acquisition time for a single spectrum was typically 10 s for three successive acquisitions at each location. A spectral resolution of 1 cm^−1^ was achieved by concurrently collecting (at each measurement) an internal reference signal from a selected neon lamp to calibrate the spectrometer. Raman spectra were collected at each investigated location of each tibia sample. An average of 6 spectra for each of the three locations and experimental condition on rat tibias were analyzed in order to obtain statistically representative spectra.

The laser spot size in the focal plane under the above-selected experimental conditions was ~5 μm in diameter. The extent of laser penetration depth along the subsurface of the rat tibia was also assessed by means of a Raman probe calibration based on in-depth defocusing of the optical focal plane. This assessment, which is shown later in detail, served to explore the possibility of non-destructively detecting and evaluating both level of infection and tissue immunochemical response.

Experimental Raman spectra were subjected to polynomial baseline subtraction and deconvolution into a series of Gaussian–Lorentzian sub-band components. The baseline subtraction was performed at once on the whole wavenumber interval investigated according to the asymmetric least square method [121]. Average spectra were deconvoluted into a series of Gaussian–Lorentzian sub-bands using commercial software (LabSpec 4.02, Horiba/Jobin-Yvon, Kyoto, Japan) according to fixed criteria for all collected spectra. All baseline-subtracted spectra were analyzed after intensity normalization to the strongest signal in each spectral interval examined. Detailed descriptions of spectral deconvolution criteria have been reported in previously published papers [13,122]. An automatic solver exploiting a linear polynomial expression of Gaussian–Lorentzian functions was iteratively run to match average experimental spectra for minimum scatter (better than 95% confidence interval) with the experimental (average) spectrum.

### 4.3. Statistical Analyses

The statistical relevance of the parameters extracted from the Raman experiments was analyzed by computing mean values and standard deviations over six measurements (*n* = 6) at each location of each rat tibias. Statistical validity was evaluated by applying the unpaired Student’s *t*-test. Values *p* < 10^−2^ and *p* < 10^−3^ were considered as statistically significant and labeled with two and three asterisks, respectively.

## 5. Conclusions

In this work, we used Raman spectroscopy to investigate at the molecular level the adverse effects of *S. aureus* infection on bone tissue using a model of direct bacterial injection in the tibia medullary cavity of Wistar/ST rats. The results showed a possible use of the Raman technique as a diagnostic tool for staphylococcal osteomyelitis. Raman spectra of infected bone contained straightforward information about levels of infection and immune response, the extent of osteoblast apoptosis, and the metabolic state of bacterial cells. In this work, we discussed the physiological state of both cells and bacteria and proposed a series of quantitative spectroscopic biomarkers to evaluate their interactions within bone medullary cavity. In particular, we sought to understand the relationship between the degree of immune response and bacterial degradation profile. From a clinical perspective, the implications of in situ non-invasive Raman diagnosis of osteomyelitis would lead to a paradigmatic shift in orthopedic practice, since this technique would facilitate a prompt, systematic, and efficient analysis of bone tissue, thereby contributing to saving patients from long-term morbidity and successive surgeries.

## Figures and Tables

**Figure 1 ijms-26-08572-f001:**
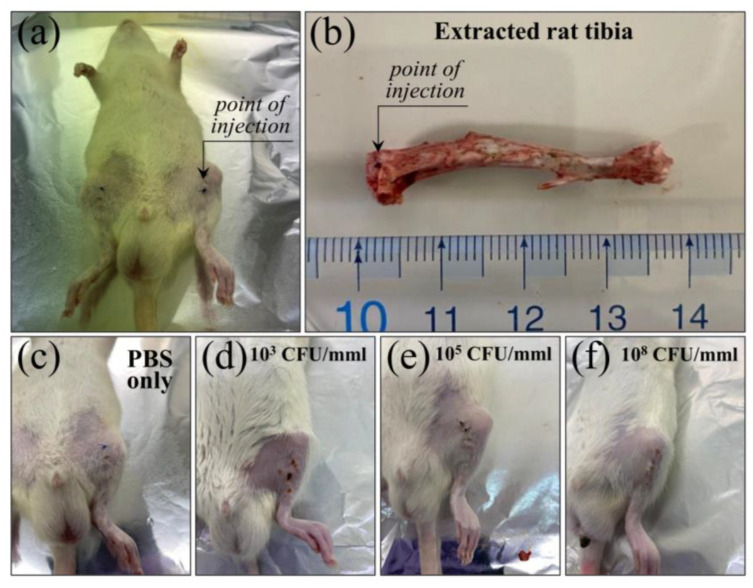
(**a**) Anesthetized rats after inoculation and (**b**) its extracted tibia after 7 d post inoculation; (**c**–**f**) pictures of rats at 7 days post injection with 100 μL PBS only, and with the same amount of PBS added with 10^3^, 10^5^, and 10^8^ CFU/mmL concentrations of *S. aureus* bacteria, respectively.

**Figure 2 ijms-26-08572-f002:**
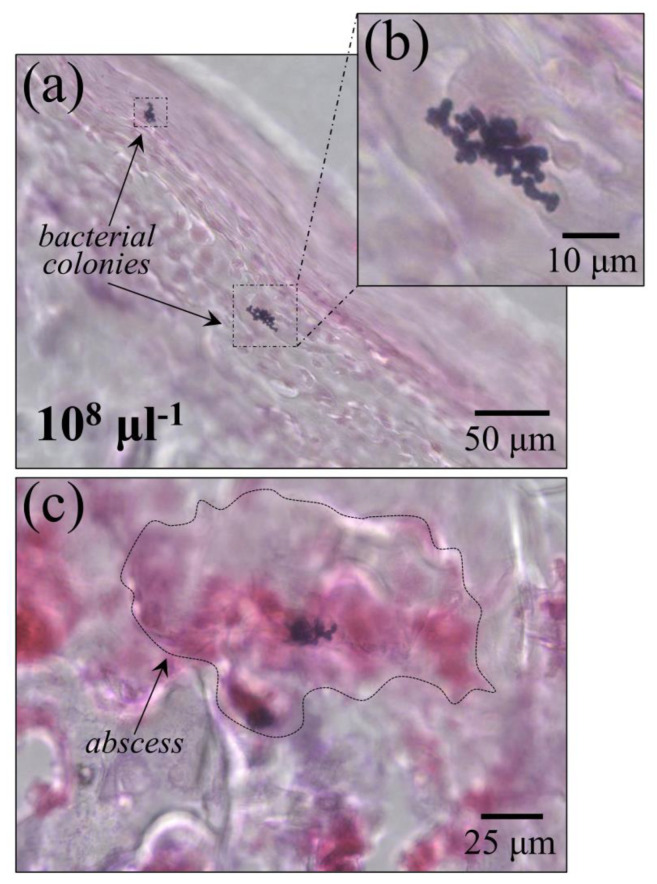
(**a**) Optical micrograph of *S. aureus* colonies as observed on a tibia cross section after bacterial inoculation at 7 d post infection (enlarged micrograph in (**b**)); in (**c**), optical micrograph showing the formation of abscesses post bacterial inoculation.

**Figure 3 ijms-26-08572-f003:**
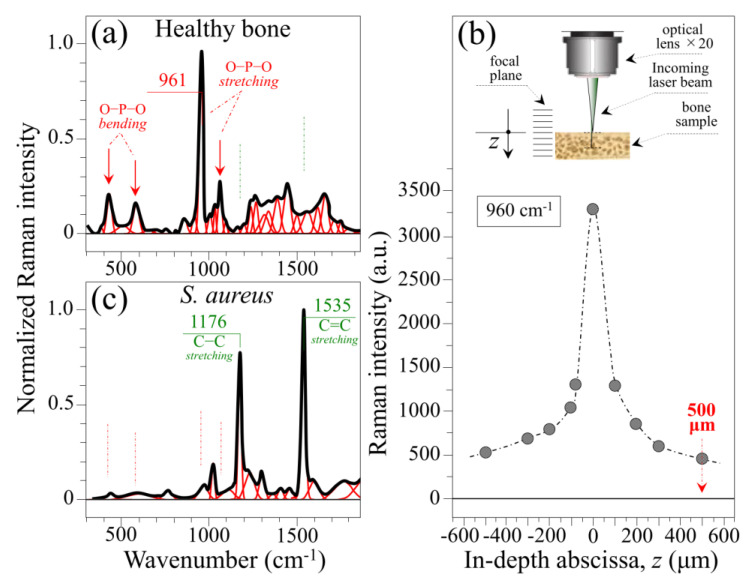
Reference Raman spectra collected on (**a**) the surface of healthy tibia and (**c**) *S. aureus* bacteria cultured in vitro (wavenumbers in cm^−1^ and vibrational assignments in inset); (**b**) scattering intensity calibration of the 961 cm^−1^ Raman band of bone hydroxyapatite to establish the penetration depth of the Raman probe (cf. draft in inset showing the shift in the focal plane above and below the surface of the bone sample).

**Figure 4 ijms-26-08572-f004:**
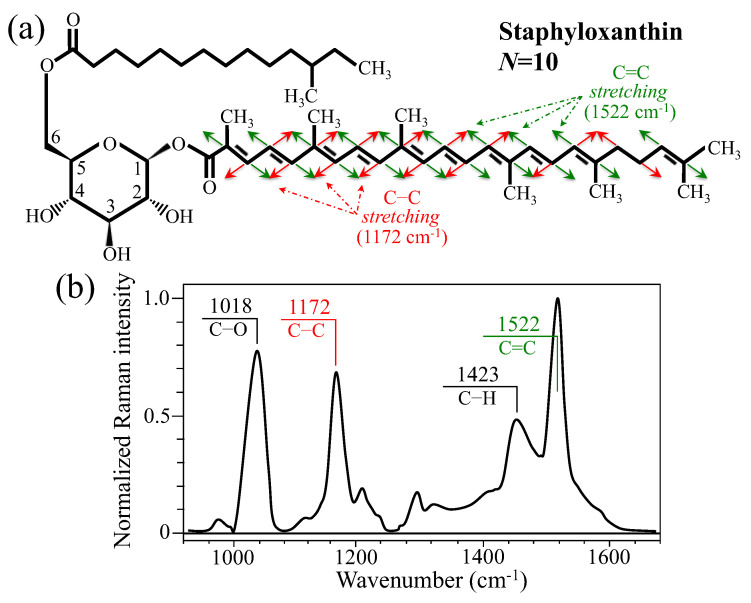
Structure (**a**) and Raman spectrum of pure staphyloxanthin compound (from Ref. [30]); vibrational modes of inset are emphasized in (**a**) and their respective signals labeled in (**b**). Red and green arrows represent C–C and C=C bond stretching, respectively (cf. corresponding wavenumbers in inset).

**Figure 5 ijms-26-08572-f005:**
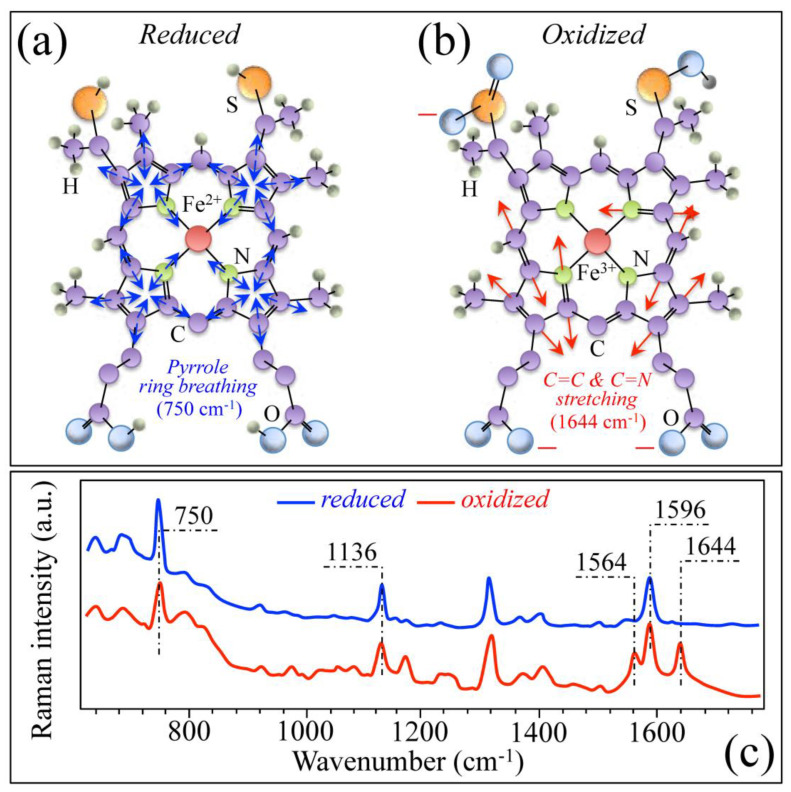
Structures of cytochrome *c* in (**a**) reduced and (**b**) oxidized states; their respective Raman spectra (cf. labels in inset) in the wavenumber interval 600~1750 cm^−1^ (from Ref. [34]) are displayed in (**c**) (wavenumbers in inset are in cm^−1^ units). Blue arrows in (**a**) and red arrows in (**b**) represent pyrrole ring breathing and C=C/C=N bond stretching, respectively (cf. corresponding wavenumbers in inset).

**Figure 6 ijms-26-08572-f006:**
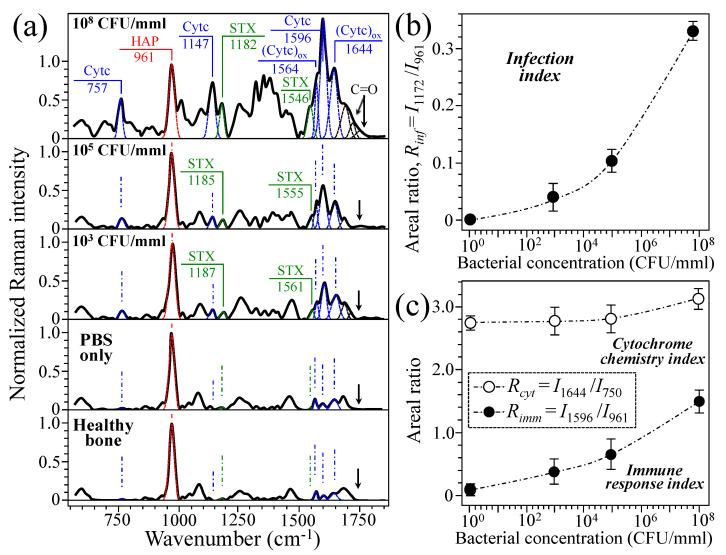
(**a**) Average Raman spectra collected in the neighborhood of the inoculation point of rat tibias immediately after their extraction at 7 d post bacterial injection: from top to bottom, samples inoculated with 100 μL PBS containing 10^8^, 10^5^, 10^3^ CFU/mmL bacterial concentrations, samples inoculated with only 100 μL PBS, and non-inoculated healthy bone as control samples (number of spectral measurements, *n* = 6, on each rat sample; cf. labels in inset); main signals belonging to hydroxyapatite (HAP), staphyloxanthin (STX), and cytochrome *c* (Cytc) are labeled in inset. Given the relatively large volume of the Raman probe and the very close locations at which spectra were recorded for each spectral condition, spectral scatter remained confined below ±0.02 in relative intensity at any wavenumber. In (**b**), a plot of the areal ratio, *R_inf_
*= *I*_1172_/*I*_961_, is given as a function of initial bacterial concentration, while plots related to cytochrome chemistry and immune response, represented by the areal ratios, *R_imm_
*= *I*_1596_/*I*_961_ and *R_cyt_
*= *I*_1644_/*I*_750_, respectively, are given in (**c**) as functions of initial bacterial concentration. An examination of the plot in (**b**) suggests that the lower threshold for *R_imm_* under the present conditions for Raman signal acquisition should lie at around 10^2^ CFU/mmL. Note that the wavenumbers in the subscript of the intensity ratios are given as those of the reference molecules in Figure 3, Figure 4 and Figure 5, since they shift in measurements on bone tissue for the reasons explained in the text. In each plot, error bars indicate standard deviations at each data point.

**Figure 7 ijms-26-08572-f007:**
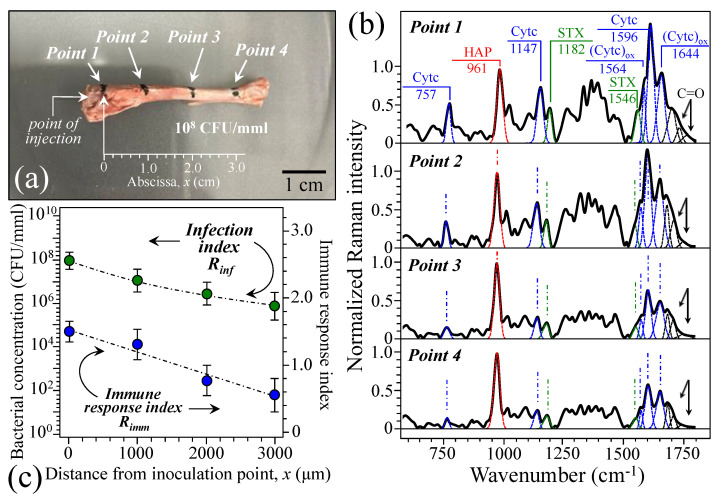
(**a**) Rat tibia initially inoculated with 100 μL PBS with a bacterial concentration of 10^8^ CFU/mmL and then extracted at 7 d post inoculation; bacterial injection at point 1, Raman measurements at points 1~4 (cf. scale and labels in insets). Spectra collected at points 1~4 are displayed from top to bottom in (**b**) (cf. main bands assigned with labels in inset; abbreviations are the same as those in Figure 6). Given the relatively large volume of the Raman probe and the very close locations at which spectra were recorded for each spectral condition, the spectral scatter remained confined below ±0.02 in relative intensity at any wavenumber. (**c**) ratios *R_inf_* and *R_imm_* as computed at different locations and representative of bacterial concentration and immune response in bone tissue, respectively (error bars indicate standard deviations at each data point).

**Figure 8 ijms-26-08572-f008:**
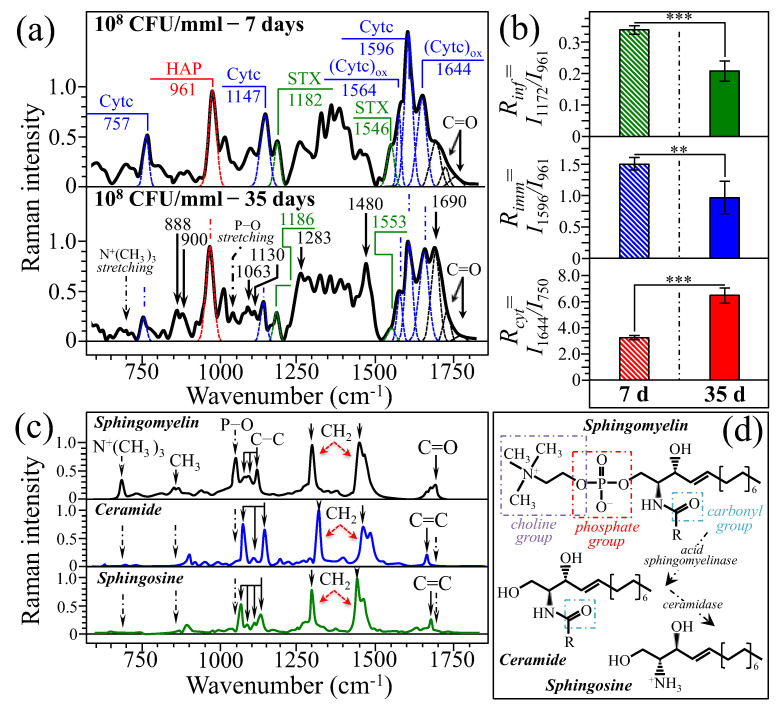
(**a**) Comparison between average Raman spectra collected on rat tibias inoculated with 100 PBS–10^8^ CFU mmL^−1^ bacterial concentration and then extracted after 7 and 35 d post inoculation (upper and lower spectrum, respectively; abbreviations are the same as those in Figure 6 and Figure 7); given the relatively large volume of the Raman probe and the very close locations at which spectra were recorded for each spectral condition, the spectral scatter remained confined below ±0.02 in relative intensity at any wavenumber. In (**b**), comparison between spectroscopic parameters *R_inf_*, *R_imm_*, and *R_cyt_* at 7 and 35 d post bacterial inoculation (asterisks give the statistical relevance of data for *n* = 6 measurements for each rat sample, as explained in Section 4.3). Error bars represent standard deviations at each data point. In (**c**), Raman spectra are given for sphingomyelin, ceramide, and sphingosine elementary molecules (from Refs. [44,45]; cf. vibrational assignments in inset), while (**d**) schematically shows the metabolic cycle in neutrophils that leads to the formation of ceramide and sphingosine from sphingomyelin, as regulated by enzymatic activities of sphingomyelin synthase and ceramide kinase, respectively.

**Figure 9 ijms-26-08572-f009:**
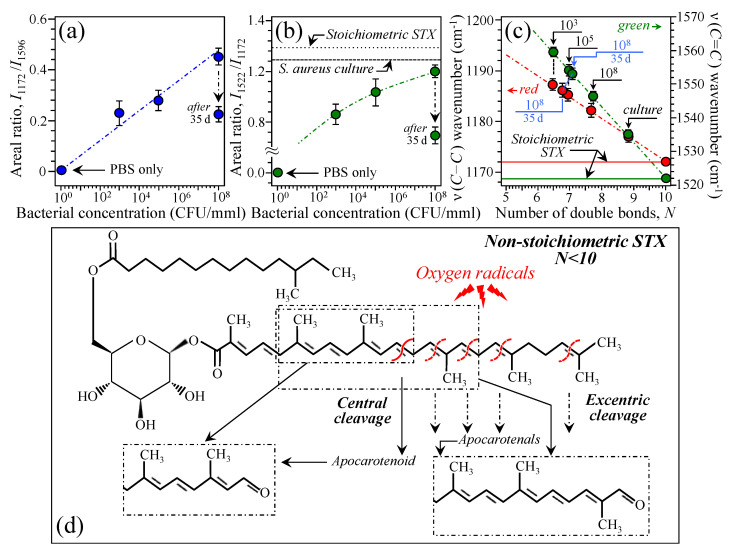
Plots of spectral ratios *I*_1172_/*I*_1596_ (**a**) and *I*_1522_/*I*_1172_ (**b**) as function of initial bacterial concentration for samples at 7 d post inoculation; the plot in (**c**) features C–C and C=C stretching wavenumbers as functions of initial bacterial doses (cf. labels in inset) plotted against the average number, *N*, of C=C bonds present in the chain of staphyloxanthin molecules, according to calibrations reported by Schaffer et al. [108] (values for stoichiometric staphyloxanthin are plotted as reference). In each plot, error bars indicate standard deviations at each data point. The draft in (**d**) shows a schematic draft for the possible mechanisms of interaction between the carotenoid chain of staphyloxanthin and the free oxygen radicals produced by apoptotic osteoblasts and neutrophils (according to Ref. [111]).

## Data Availability

Data are contained within the article.

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
