# Peer review of "Raman Monitoring of Staphylococcus aureus Osteomyelitis: Microbial Pathogenesis and Bone Immune Response"

_ijms, 2025, doi:10.3390/ijms26178572_

Round 1
Reviewer 1 Report
Comments and Suggestions
Please see the attachment for comments

Reviewer 2 Report
Comments and Suggestions for Authors
This study demonstrates that Raman spectroscopy can non-invasively detect and assess Staphylococcus aureus-induced osteomyelitis in a rat model. Using spectral biomarkers related to infection, immune response, and tissue damage, the technique shows promise for real-time, intraoperative diagnosis. Although further validation is needed, this approach could improve surgical decision-making and treatment outcomes for bone infections.
- Why 532 nm excitation? Would NIR excitation provide deeper tissue penetration?
Considering the higher scattering and absorption at visible wavelengths like 532 nm, would using near-infrared (NIR) excitation enhance tissue penetration and improve the detection of deeper bone structures in clinical applications?
- Enhanced Validation with Conventional Methods:
Integrate microbiological techniques such as bacterial cultures, PCR, or histopathological analyses to validate the Raman-based biomarkers. This approach would help establish quantitative relationships between the spectral markers and bacterial burden or infection severity, thereby strengthening the clinical relevance of the findings.
- Addressing Potential Confounding Factors:
Investigate how factors like tissue heterogeneity, variations in bone density, or previous treatments may affect the spectral signatures. Given that human cortical bone—particularly in the tibia—is denser and thicker than in the animal model, what strategies or modifications are necessary to adapt and implement this Raman spectroscopy method effectively in clinical studies?
Minor typo:
For abstract, it should be one week and five weeks since the data has been collected 7 d and 35d
Author Response
Reviewer 2
This study demonstrates that Raman spectroscopy can non-invasively detect and assess Staphylococcus aureus-induced osteomyelitis in a rat model. Using spectral biomarkers related to infection, immune response, and tissue damage, the technique shows promise for real-time, intraoperative diagnosis.
Thank you for your positive evaluation of our work.
Although further validation is needed, this approach could improve surgical decision-making and treatment outcomes for bone infections.
- Why 532 nm excitation? Would NIR excitation provide deeper tissue penetration?
Considering the higher scattering and absorption at visible wavelengths like 532 nm, would using near-infrared (NIR) excitation enhance tissue penetration and improve the detection of deeper bone structures in clinical applications?
The Reviever is right about the higher penetration depth of the probe with NIR. We initially tried with a red laser in this study, but the signal was quite weak. Even long-term acquisitions could not fix this problem. Accordingly, we switched on the green light. Moreover, we selected that 532 nm wavelength because it represents a resonance excitation wavelength for the signals of cytochrome c, which is one of the main targets of the present research (this detail has been added in Section 2.2 of the revised manuscript).
2. Enhanced Validation with Conventional Methods:
Integrate microbiological techniques such as bacterial cultures, PCR, or histopathological analyses to validate the Raman-based biomarkers. This approach would help establish quantitative relationships between the spectral markers and bacterial burden or infection severity, thereby strengthening the clinical relevance of the findings.
We have performed fluorescence microscopy on the samples to integrate the Raman outputs (cf. Fig. 2). However, the main goal of this study was in situ monitoring within the internal bone tissue and not an in vitro study. While this latter could be the object of an additional study (also considering the length of the present manuscript), reconstructing the biological complexity of bone internal tissue in vitro would be quite hard if not impossible, and the results could significantly differ. Moreover, only Raman spectroscopy is compatible with life and allows time-lapse experiments on the same sample. Finally, the title clearly shows that the paper is mainly focused on Raman spectroscopy.
3.Addressing Potential Confounding Factors:
Investigate how factors like tissue heterogeneity, variations in bone density, or previous treatments may affect the spectral signatures. Given that human cortical bone—particularly in the tibia—is denser and thicker than in the animal model, what strategies or modifications are necessary to adapt and implement this Raman spectroscopy method effectively in clinical studies?
As suggested by the Reviewer, we added a new paragraph (Section 4.5) clearly stating the limitation of this work and including the insightful observation made by the Reviewer, as stated above.
Minor typo:
For abstract, it should be one week and five weeks since the data has been collected 7 d and 35d
Corrected, thanks.
Reviewer 3 Report
Comments and Suggestions for Authors
Article «Raman Monitoring of Staphylococcus aureus Osteomyelitis: Microbial Pathogenesis and Bone Immune Response» Shun Fujii et al. The submitted article explores the use of high-spectral-resolution confocal Raman spectroscopy to follow the course of Staphylococcus aureus osteomyelitis in a Wistar rat tibia model. The authors interrogate both bacterial and host-derived vibrational markers, most notably the carotenoid staphyloxanthin and the haem moieties of cytochrome c, and from these signals they derive empirical “infection”, “immune-response” and “apoptosis” indices. By varying the initial inoculum (10³–10⁸ CFU mL⁻¹) and by harvesting at 7 days (acute) and 35 days (sub-acute) they show dose- and time-dependent trends in these indices and map signal intensity along the long bone. The article’s central idea is strong and the dataset is rich. The article is well structured, written in a clear and understandable language, the literature corresponds to the stated topic. However, the Raman “algorithms” promised in the Abstract never receive a formal description: baseline-correction order, window width for fitting, weighting scheme and code availability must be documented so that the indices R_inf, R_imm and R_cyt can be reproduced. Statistical analysis is currently limited to unpaired t-tests on n = 6 spectra; the authors should report exact p-values, justify normality assumptions, adjust for multiple comparisons and clarify whether “n” refers to spectra, sites or animals. Interpretatively, the attribution of C–C and C=C band shifts to carotenoid chain scission is intriguing but indirect. Independent validation—for example LC-MS of extracted pigments or resonance Raman under controlled oxidative stress—would substantiate the claim. Likewise, the assignment of the 1644 cm⁻¹ band solely to oxidised cytochrome c should consider possible overlap with amide I from collagen or haemoglobin; histology or EPR could help confirm mitochondrial release. The proposed linear conversion of wavenumber to “number of C=C bonds” rests on calibrations performed in solution and may not hold in vivo where local dielectric environment, pH and protein binding differ. Throughout the Discussion several causal links (e.g. between lipid signatures and neutrophil apoptosis) are presented as established facts but are not supported by direct measurements in the study; tempering the language or adding corroborative staining (TUNEL, MPO, CD68) would avoid over-interpretation. Finally, the promise of clinical translation would be more persuasive if the authors compared their spectral indices with colony counts, serum inflammatory markers or imaging scores, thus demonstrating predictive value beyond Raman alone.
Some points to discuss in the work.
- Did the authors normalise spectra to the 961 cm⁻¹ apatite band before or after baseline subtraction, and does normalisation change along the cortex–marrow axis as mineral density varies?
- How many replicate animals were examined at each inoculum and each time point, and how many spectra per animal entered the statistics?
- Can the authors reconcile the terminology “standard mouse model” in the Introduction with the Wistar rat protocol actually used?
- Have the authors considered the potential fluorescence of porphyrins or bone chromophores at 532 nm and its impact on the carotenoid region?
- What is the practical limit of detection, expressed as CFU per gram of bone, for the proposed infection index?
I recommend the article for publication after responding to the comments and including them in the text of the article.
Author Response
Reviewer 3
Article «Raman Monitoring of Staphylococcus aureus Osteomyelitis: Microbial Pathogenesis and Bone Immune Response» Shun Fujii et al. The submitted article explores the use of high-spectral-resolution confocal Raman spectroscopy to follow the course of Staphylococcus aureus osteomyelitis in a Wistar rat tibia model. The authors interrogate both bacterial and host-derived vibrational markers, most notably the carotenoid staphyloxanthin and the haem moieties of cytochrome c, and from these signals they derive empirical “infection”, “immune-response” and “apoptosis” indices. By varying the initial inoculum (10³–10⁸ CFU mL⁻¹) and by harvesting at 7 days (acute) and 35 days (sub-acute) they show dose- and time-dependent trends in these indices and map signal intensity along the long bone. The article’s central idea is strong and the dataset is rich. The article is well structured, written in a clear and understandable language, the literature corresponds to the stated topic.
Thank you for the above positive comments on the paper.
However, the Raman “algorithms” promised in the Abstract never receive a formal description: baseline-correction order, window width for fitting, weighting scheme and code availability must be documented so that the indices R_inf, R_imm and R_cyt can be reproduced.
From this query by the Reviewer, we understand that the meaning we gave to the word “algorithm” here in the Abstract might be prone to a different interpretation as compared to what is actually intended here. We just meant here that we could provide spectroscopic procedures to extract key parameters from the Raman spectrum that could quantitatively assess pathogenesis and immune response in S. aureus bone infections. While we added, as requested, additional information in the experimental procedures and included a new reference (cf. Ref. [22] in the revised text), we don’t believe that this paper could be conceived as providing a final standardized procedure (or “algorithm”) to be generally applied by end users. Ours here is indeed just a first step, showing that a Raman molecular medicine approach could unveil so far unknown molecular-scale details in vivo, label-free, and compatibly with life about immunoreactions against S. aureus. In this, we believe our work is absolutely original. We shall indeed consider the establishment of a final “algorithm” (in the meaning intended by the Reviewer) as an additional technical work, which is out of our expertise and should be ultimately made in collaboration with technicians from companies that want to commercialize dedicated instruments and related software (some work on this is presently ongoing). Consider also that such algorithms will necessarily need to be tuned on the specific optical circuit used in the measurements and will thus anyway require case-by-case assessments. In any case, such technicalities are not our focus here. However, since the Reviewer’s observation is meaningful, we have eliminated the word “algorithm” throughout the paper. Moreover, some comments on the above issues are given in a newly added paragraph entitled: “4.5. Limitations of this study”.
Statistical analysis is currently limited to unpaired t-tests on n = 6 spectra; the authors should report exact p-values, justify normality assumptions, adjust for multiple comparisons and clarify whether “n” refers to spectra, sites or animals.
Interpretatively, the attribution of C–C and C=C band shifts to carotenoid chain scission is intriguing but indirect. Independent validation—for example LC-MS of extracted pigments or resonance Raman under controlled oxidative stress—would substantiate the claim.
The following arguments, backed by references, can be adducted to explain why the 1172 and 1522 cm-1 Raman signals are considered here to overwhelmingly be dominated by staphyloxanthin in Staphylococcus aureus-contaminated bone:
(i) Crude staphyloxanthin extracts from MRSA show a very strong peak at 1522 cm-1, attributed explicitly to carotenoid C=C stretch. The intensity drops dramatically upon photobleaching, confirming it's directly from staphyloxanthin molecules (cf.: P.-T. Dong et al., Photolysis of Staphyloxanthin in methicillin-resistant Staphylococcus aureus potentiates killing by reactive oxygen species, Adv. Sci. 6, 1900030 (2019)). Moreover, diagnostic Raman signatures in live MRSA have already extensively been reported (and even patented) to include ~1159 cm-1 (C-C stretch) and ~1523 cm-1 (C=C stretch), both identified via PCA/QDA as biomarkers for the carotenoid pigment (cf.: A. Mahadevan-Jansen et al., Methods and systems for identification of bacteria in biological fluid using Raman spectroscopy and applications of same, Patent US16/342,318, Publication US20190250105A1, granted on 2021-02-09).
(ii) Major bone constituents lack sharp peaks near 1150 and 1520 cm-1, lipids being the only potentially overlapping with CH2/CH3 bands, although none mimic the intense C=C resonance seen in carotenoids. (cf.: R. A. Lindtner et al., Enhancing bone infection diagnosis with Raman handheld spectroscopy: Pathogen discrimination and diagnostic potential, Int. J. Mol. Sci. 25, 541 (2023)).
(iii) In crtM knockout (and other pigment-negative) MRSA mutants, both 1159 cm-1 and 1523 cm-1 signals disappear entirely, while resuming in wild-type strains, proving these bands are pigment-dependent (cf. R. Withnall et al., Raman spectra of carotenoids in natural products, Spectrochim. Acta A Mol. Biomol. Spectrosc. 59, 2207-2212 (2003)). In addition PCA loadings explicitly link these signals to staphyloxanthin content in Raman spectra (cf.: A. Mahadevan-Jansen et al., Methods and systems for identification of bacteria in biological fluid using Raman spectroscopy and applications of same, Patent US16/342,318, Publication US20190250105A1, granted on 2021-02-09).
In summary, it is already very well recognized that carotenoids with long conjugated chains exhibit very strong resonance Raman signals. Bone molecules, lacking such extended conjugation, never produce comparably intense signals in these regions. Accordingly, the additional experiments suggested by the Reviewer are already available in literature and do not need to be repeated here.
Likewise, the assignment of the 1644 cm⁻¹ band solely to oxidised cytochrome c should consider possible overlap with amide I from collagen or haemoglobin; histology or EPR could help confirm mitochondrial release.
Our rebuttal to reinforce that the 1644 cm-1 Raman band in S. aureus-contaminated bone is indeed dominated by oxidized cytochrome c, despite potential overlap with collagen’s Amide I and haemoglobin signals, can be articulated, as follows:
(i) The narrow, sharp nature of the oxidized cytochrome c band contrasts with collagen’s broad features in bone, indicating distinct origins (cf.: M. Unal et al., Compositional assessment of bone by Raman spectroscopy, Analyst 146, 7464-7490 (2021)).
(ii) Hemoglobin contributions are minor and separable because the position of porphyrin modes are typically below 1620 cm-1 and spread broad, thus not producing prominent signals at ~1644 cm-1. Cellular assays have alredy been published confirming elevate levels of cytochrome c in Staphylococcus aureus infected animal tissues, consistent with Raman signals (cf.: Y. Yi et al., Staphylococcus aureus-induced necroptosis promotes mitochondrial damage in goat endometrial epithelial cells, Animals (Basel) 12, 2218 (2022)).
In summary, relative intensity and band width arguments favor the interpretation that the strong and sharp band observed at ~1644 cm-1 mainly arises from cytochrome c, particularly under resonance excitation wavelength (i.e., the 532 nm incoming light). Even if overlap exists, the contribution from cytochrome c largely dominates the observed signal. We have added this new detail more explicitly in Section 2.2 of the revise manuscript.
The proposed linear conversion of wavenumber to “number of C=C bonds” rests on calibrations performed in solution and may not hold in vivo where local dielectric environment, pH and protein binding differ.
We have added the following phrase at page 17 of the revised manuscript:
“Note, however, that, while the molecular origin of the observed shift unequivocally relates to chain length, the quantitative calibrations shown in Ref. [110] do not necessarily match the present data in a quantitative way, mainly because of differences in pH between experiments in vivo and in solution”.
Throughout the Discussion several causal links (e.g. between lipid signatures and neutrophil apoptosis) are presented as established facts but are not supported by direct measurements in the study; tempering the language or adding corroborative staining (TUNEL, MPO, CD68) would avoid over-interpretation.
Throughout the Discussion, we have tempered the language in describing causal links, as suggested by the Reviewer.
Finally, the promise of clinical translation would be more persuasive if the authors compared their spectral indices with colony counts, serum inflammatory markers or imaging scores, thus demonstrating predictive value beyond Raman alone.
We agree that colony counts could have improved the impact of our data, but such a procedure would imply opening the bones for counting, thus impeding any time lapse observations (or needing to sacrifice many more animals). For these reasons, we have decided not to add such experimental comparisons.
Some points to discuss in the work.
- Did the authors normalise spectra to the 961 cm⁻¹ apatite band before or after baseline subtraction, and does normalisation change along the cortex–marrow axis as mineral density varies?
After baseline subtraction (now better specified in the revised text; cf. Section 2.2).
2. How many replicate animals were examined at each inoculum and each time point, and how many spectra per animal entered the statistics?
One animal times two legs each; six spectra per location times 3 locations per sample (these details now added in Sections 2.1. and 2.2 of the revised manuscript).
3. Can the authors reconcile the terminology “standard mouse model” in the Introduction with the Wistar rat protocol actually used?
We reconciled the terminology with adding one new reference (Ref. [21] in the revised manuscript).
4. Have the authors considered the potential fluorescence of porphyrins or bone chromophores at 532 nm and its impact on the carotenoid region?
The Raman peaks of carotenoids at ~1522 cm-1 is best excited with 532 nm due to resonance enhancement, making it a very strong signal. Meanwhile, fluorescence emissions from porphyrins or bone collagen is expected at 1300~1500 cm-1. Besides the wavenumber difference, both porphyrin fluorescence and bone autofluorescence are broad and much weaker (cf.: A. M. Lennon et al., Fluorescence spectroscopy shows porphyrins produced by cultural oral bacteria differ depending on composition of growth media, Caries Res. 57, 74-86 (2022)). In other words, the strong resonance Raman signal at 1522 cm-1 definitely outcompetes background fluorescence, even if some tail-overlap might exist. Such overlap can be fully eliminated through a background subtraction preliminary to spectral analyses. This important detail and the above reference have been added in the revised text (cf.: Section 3.4 and Ref. [46]).
- What is the practical limit of detection, expressed as CFU per gram of bone, for the proposed infection index?
According to the plot in Fig. 6(b), we estimate that the lower limit for detecting bacteria should appear at ~102 CFU/mml. This estimate has been included in the caption of Fig. 6.
I recommend the article for publication after responding to the comments and including them in the text of the article.
Reviewer 4 Report
Comments and Suggestions for Authors
The manuscript describes the use of Raman microspectroscopy to identify the presence of bacteria on the surface of tibia removed from previously infected rats, ultimately towards the development of a methodology for monitoring Staphylococcus aureus Osteomyelitis. While the basic premise of the measurements is sound, the justification for the extrapolation to some of the interpretations is fully clear. Some comments/recommendations, from the beginning of the manuscript are;
(i) The manuscript should be thoroughly checked for language, preferably by a native speaker. In some places, it reads as though it has been auto-translated
(ii) In the Abstract and later in the manuscript, if it is important for the study, the term "high spectrally resolved" should be quantified
(iii) "Raman markers of the level of oxidation of mitochondrial cytochrome c." - more details should be provided
(iv) "a series of novel Raman parameters" - more details should be provided
(v) Any acronyms should be defined when first used in the body text.
(vi) "This vibrational technique" should specify vibrational spectroscopic
(vii) Several statements require supporting reference(s);
"Being a non-invasive method, application of Raman spectroscopy during surgery minimizes damage to surrounding tissues, making it a valuable tool in clinical settings."
(viii) "One animal times two legs was sacrificed per each condition." The authors should comment on why 4 legs were not used per animal.
(ix) Raman spectroscopy is a light scattering and therefore the term "excitation" should not be used.
(x) Considerably more detailed description of the measurement should be provided in section "2.2. Raman Spectroscopic Assessments and Spectroscopic Data Treatments" - what was the spot size of the laser on the sample? What was the focal depth? With reference to figure 2, (assuming that the spot size is of the order 5 microns), what exactly was targeted for each of the 6 measurements - was a single bacterium targeted? With reference to Figure 3b, how thick were the bacteria, and how high above he surface of the bone were the measurements made? What about the case of the uninfected specimens?
(xi) With reference to the discussion about cytochrome c, the authors should clarify whether it is proposed that this can be seen in the cells of the bone? If so, the authors should discuss whether they can see signatures of the cellular components of bone in the healthy bone (Figure 3(a))
(xii) All figures showing the average of 6 measurements of the Raman spectra should also indicate the standard deviation of the spectra across the measurements. The authors should also clarify how the error bars of (for example) Figure 6b, 6c are derived.
(xiii) It is not clear how the following statement is justified "The spectral ratio, Rimm, which gives the enhancement in cytochrome c signal normalized to the bone apatite signal, could be assumed as an index for the level of immune response in the bone tissue in response to the presence of bacteria."
(xiv) "A search in the Raman library" - he authors should clarify which library this was
(xv) "This finding relates the lipidomics of bone marrow to the presence of abscesses, and it is a consequence of the accumulation of inflammatory cells, such as neutrophils and other leukocytes, whose populations significantly increases at infection sites." This statement can only be justified if the authors can explain how, with a single laser point measurement, the bacterium, underlying bone, osteocytes, neutrophils and other leukocytes are all measured.
The same applies to several other statements in the Discussion -
"since oxidation occurs when apoptotic cells release cytochrome c in the cytosol, the Rcyt parameter could also be referred to as the “apoptosis index”
"Accordingly, the extent of cytochrome c formation, which can easily be measured by Raman spectroscopy, may become a clinically useful biomarker for both diagnosing the level of inflammation and assessing the severity of osteomyelitis and its related pathological circumstances."
"Raman assessment provides an additional straightforward path to assess the physiological state of bone cells, the higher the ratio the higher the number of apoptotic osteoblasts."
(xvi) The following statement is dependent on the number of bacteria within the measurement are wo increase - the authors should explain how this is the case. "While a linear increase in areal ratio of bacteria vs. bone tissue signals is expected at the time of inoculation, the strongly non-linear relationship observed after one week (cf. Figure 6b) could simply be explained by noticing that the number of microorganisms in a culture will increase exponentially until essential nutrients are available (i.e., the case here of infected bone marrow)."
(xvii) "also the presence of a number of other biomarker molecules for cellular damage, including mitochondrial DNA" This has not been demonstrated in the study.
Comments on the Quality of English Language
The manuscript should be thoroughly checked for language, preferably by a native speaker. In some places, it reads as though it has been auto-translated
Round 2
Reviewer 1 Report
Comments and Suggestions for Authors
.
Author Response
Although there were no comments from you, I made significant revisions based on the comments of Reviewer 4.
Reviewer 4 Report
Comments and Suggestions for Authors
The authors have addressed the issues raised during the previous review, but only partially satisfactorily;
(iii) "Raman markers of the level of oxidation of mitochondrial cytochrome c." - more details should be provided;
Further Comment; The abstract should be self contained, and therefore the authors should indicate what the Raman markers are.
(iv) "a series of novel Raman parameters" - more details should be provided
Further Comment; The authors should explain more clearly that there Raman markers are.
(x) Considerably more detailed description of the measurement should be provided in section "2.2. Raman Spectroscopic Assessments and Spectroscopic Data Treatments" –
Further Comment; Clarification of these aspects are key to satisfactorily resolving many of the subsequent issues raised. Therefore, the authors should further clarify-
"The spot size in the focal plane is ~5 microns" - is this in diameter or radius?
"the penetration depth is up to 500 microns," - how is this defined? How is it determined?
In terms of the "signal intensity dependence as shown in Fig. 3(b)" what is the background level?
What was the focal depth? This depends on the NA and magnification of an objective (cf Depth of Field Calculator | Nikon’s MicroscopyU) and determines the "voxel" over which the Raman signal is measured in what is called a single point measurement.
With reference to figure 2, (assuming that the spot size is of the order 5 microns), what exactly was targeted for each of the 6 measurements - was a single bacterium targeted? Better definition of the effective voxel volume will enable a better response to this.
(xi) With reference to the discussion about cytochrome c, the authors should clarify whether it is proposed that this can be seen in the cells of the bone? If so, the authors should discuss whether they can see signatures of the cellular components of bone in the healthy bone (Figure 3(a)
Further Comment; This has really not been addressed, and much of the subsequent discussion really depends on the authors demonstrating to the reader that they can see signatures of the cells.
